# Role of Molecular Weight in Polymer Wrapping and Dispersion of MWNT in a PVDF Matrix

**DOI:** 10.3390/polym11010162

**Published:** 2019-01-17

**Authors:** Muthuraman Namasivayam, Mats R. Andersson, Joseph Shapter

**Affiliations:** 1Flinders Centre for Nanoscale Science and Technology, College of Science and Engineering, Flinders University, Bedford Park, Adelaide, South Australia 5042, Australia; nama0009@flinders.edu.au; 2Australian Institute for Bioengineering and Nanotechnology, The University of Queensland, St. Lucia, Brisbane, Queensland 4072, Australia

**Keywords:** Polyvinylpyrrolidone, Poly(4-vinylpyridine), polymer nanocomposite, thermal conductivity, polymer wrapping

## Abstract

The thermal and electrical properties of a polymer nanocomposite are highly dependent on the dispersion of the CNT filler in the polymer matrix. Non-covalent functionalisation with a PVP polymer is an excellent driving force towards an effective dispersion of MWNTs in the polymer matrix. It is shown that the PVP molecular weight plays a key role in the non-covalent functionalisation of MWNT and its effect on the thermal and electrical properties of the polymer nanocomposite is reported herein. The dispersion and crystallisation behaviour of the composite are also evaluated by a combination of scanning electron microscopy (SEM) and differential scanning calorimetry (DSC).

## 1. Introduction

Carbon nanotubes are cylindrical tubes with a π conjugated 1D structure which can be thought of as being made from graphene and have outstanding electrical, optical, thermal and mechanical properties [1]. They can also be described as cylinders formed through a hexagonal arrangement of carbon atoms. Based on their structure, nanotubes can be differentiated as single walled nanotubes (SWNT) consisting of an individual rolled up sheet of graphene or multi walled nanotubes (MWNT) consisting of several graphene cylinders bound together by weak Van der Waals forces [2]. Despite being a promising candidate for various applications, including energy conversion, electronics, sensors etc., the high aspect ratio and strong Van der Waals interactions limit their solubility and hinder their uniform dispersion in a polymer matrix [3]. 

The electrical and thermal conductivity of a polymer composite depends on both the polymer and the filler. One of the main aspects in polymer nanocomposites is the interaction between the polymer and the filler. Even though carbon nanotubes happen to be a compatible choice as a conductive filler, they fail to match the theoretical predictions [4]. This can be attributed to a number of factors, including alignment, volume fraction and most importantly, the dispersion of nanotubes in the matrix. A good dispersion of CNTs in a polymer matrix is essential to achieving high electrical and thermal conductivity. Methods like in situ polymerisation and solution processing can produce successful dispersions in a polymer matrix, but this comes at the cost of requiring CNT functionalisation, which causes damage to the structure of CNTs. Non-covalent functionalisation is a feasible process in achieving a successful dispersion of CNTs in a polymer matrix without causing significant changes in their electronic and mechanical properties. 

Carbon nanotubes are electron rich moieties and as such, can interact with any electron deficient species to form a donor-acceptor complex. Polymer wrapping is achieved through non-covalent interaction between the π-system of CNT and the functional groups contained in the polymer [5]. Recently, Eklund, Murray et al. showed that the noncovalent interactions between CNTs with small molecules using interactions other than π–π interactions, such as CH–π and cation–π, lead to adsorption. These interactions, although they form stable CNT–polymer dispersions, are comparatively weaker compared to π–π interactions [6,7,8]. Poly(vinylidene fluoride) (PVDF) is a semi crystalline polymer with different crystalline forms and exciting properties, including superior piezoelectric and pyroelectric properties. Because of its excellent mechanical and electrical properties, PVDF has many important commercial and technological applications, ranging from supercapacitors, transducers, and actuators to batteries. PVDF/CNT nanocomposites have drawn considerable attention due to their low preparation cost and solution processability. These conducting polymers open up opportunities in the field of electronics, ranging from flexible sensors and nanogenerators to super hydrophobic membranes and UF membranes for waste water management [9]. Much work towards understanding the effect of CNTs in PVDF composites has been conducted [10,11]. G.H. Kim and S.M. Hong investigated the relationship between the structure and physical properties of s1a PVDF/MWNT blend and concluded that permittivity, electrical conductivity and thermal conductivity increase with increasing MWNT content. However, a critical conductivity saturation point and percolation threshold for the PVDF/MWNT composite are also observed, indicating that any further incorporation of MWNT does not alter the electrical and thermal conductivity [12]. Likewise, A. Mandal and A. K. Nandi prepared a PMMA functionalised MWNT/PVDF composite using nitrene chemistry and found that an increase in thermal stability and storage modulus is witnessed with increasing f-MWNT concentration [13]. F.-P. Du et al. demonstrated a facile method to prepare porous PVDF/MWNTs composite films with an improvement in thermoelectric properties [14]. However, to the best of our knowledge, no work has been carried out to investigate the effect of the functionalisation polymer’s molecular weight on the overall thermal conductivity and electrical conductivity of polymer composites.

In this study, a polymer nanocomposite of polyvinylpyrrolidone (PVP) functionalised MWNT in a PVDF matrix is formed. The concentration of PVDF and MWNT is held constant throughout the experiment, while the concentration of PVP is varied. Through this study, we are exploring changes in PVDF-MWNT interactions with changes in PVP molecular weight. Secondly, a comparative thermal conductivity result of a PVP wrapped MWNT/PVDF composite and poly(4-vinylpyridine) (P4VP) wrapped MWNT/PVDF composite with similar molecular weights is studied.

## 2. Materials and Methods

All the carbon nanotubes used in this research were MWNTs purchased from Sigma Aldrich (Product No.:755133, St. Louis, MO, USA), with an average diameter of 9.5 nm, a length of 1.5 μm and an impurity of less than 5% metal oxide. PVDF with a melt flow rate (MFR) of 20–35 g per 10 min (230°C, 3.8 Kg^−1^), density of 1.78 g·mL^−1^ at 25 °C and an average molecular weight of 180,000 g·mol^−1^ (Sigma Aldrich) was used as the polymer matrix. Polyvinylpyrrolidone with molecular weights of 10,000 g·mol^−1^, 40,000 g·mol^−1^, and 55,000 g·mol^−1^ (Sigma Aldrich), and poly(4-vinylpyridine) with a molecular weight 60,000 g·mol^−1^ (Sigma-Aldrich), were used for non–covalent functionalisation of nanotubes.

### 2.1. Sample Preparation 

The nanocomposites were prepared through a solution mixing method, which included two steps. The first step was to obtain a “stable solution” containing PVP or P4VP wrapped MWNTs containing the structures shown in Figure 1. The second step was to mix the “solution” and the PVDF polymer in the same solvent, followed by a controlled evaporation process through deposition over a glass or silicon substrate, resulting in a film thickness of approximately 40 μm. An Elmasonic S30H (Singen, Germany) 280 W power bath sonicator was used for the sample preparation.

First step: 1 mg of unmodified MWNT was suspended in 250 μL of DMF and sonicated for 10 min. Then, a defined amount of PVP or P4VP polymer was dissolved in the CNT dispersion to create a stable solution. The mixture was sonicated for a period of 45 min and then left undisturbed overnight. 

Second step: The mixture was mixed with 20 mg of PVDF polymer in 200 μL of DMF and sonicated for a period of 4 h. The resulting mixture of PVP@MWNT/PVDF or P4VP@MWNT/PVDF was then deposited on a clean 4 cm^2^ silicon wafer, and dried in an oven at a temperature at 100 °C for 24 h. 

In this work, the concentration of MWNT and PVDF was essentially kept constant, with only the concentration of PVP varying at a weight percent of 1.48% to 41.18% (0.025 to 0.7mg) with respect to the amount of MWNT used to make the composite. A comparative result was produced.

### 2.2. Characterisation

Nanocomposites were deposited on a clean glass substrate and electrical conductivity was measured at room temperature using a four-point probe. 

Thermal conductivity was measured using a steady state technique in a well-insulated chamber, where the sample was placed between a heat source and a heat sink, with a known amount of heat supplied through a steady state power input using a PID Temperature controller (Ocean Controls N322, Carrum Downs, Australia). The temperature difference across a given length of the sample was measured using a differential temperature meter (Fluke 52 II, Everett, WA, USA) after a steady-state temperature distribution was acquired. The thermal conductivity of the sample was calculated using Fourier’s Law of Heat Conduction.
$$k=\frac{\mathrm{QL}}{A\Delta T}$$
where, *Q* is the amount of heat supplied through the sample, *A* is the cross-sectional area of the sample, *L* is the distance through which heat flows, and Δ*T* is the temperature difference observed.

### 2.3. Differential Scanning Calorimetry

A TA Instruments 2930 (New Castle, DE, USA) was used to investigate the crystallisation and melting behaviour of the sample. A sample of about 9 mg was first heated from a temperature of 20 to 200 °C and then maintained at 200 °C for one minute, before cooling down from 200 to 20 °C at a rate of 10 °C·min^−1^. These steps were repeated to acquire a second heating scan.

### 2.4. SEM 

The dispersion of nanotubes in the nanocomposites was characterised using an Inspect F50 SEM (FEI, Oregon, USA) at an accelerating voltage of 10 kV. Samples were imaged from both the anterior and fractured side view. 

## 3. Results and Discussion

### 3.1. Dispersion of MWNTs

Pristine MWNT and PVP@MWNT exhibit good dispersion in aqueous medium with the aid of sonication, as shown in Figure 2. However, after being placed undisturbed for 120 h, the PVP treated MWNT still exhibits a stable dispersion of MWNTs, but an apparent deposition phenomenon is observed in the pristine MWNTs within 24 h of being statically placed. This confirms that an interfacial interaction is established between PVP and MWNT, apparently improving and stabilising the dispersion of MWNT in the medium.

### 3.2. Thermal Conductivity

Three different molecular weight PVPs were used to non-covalently functionalise the surface of MWNT to prepare PVP@MWNT/PVDF nanocomposites at a range of concentrations and the thermal conductivity of each sample was measured. Figure 3 shows the thermal conductivity of polyvinylpyrrolidone (PVP) wrapped MWNT/PVDF composites. The measurement observed at 0 concentration is the thermal conductivity of the MWNT/PVDF composite without the presence of any PVP. This is the thermal conductivity of unwrapped pristine MWNT in the PVDF matrix. 

A thermal conductivity of 1.48 W·m^−1^·K^−1^ is observed for the MWNT sample prepared with no PVP wrapping, which is greater than the theoretical thermal conductivity value of a pure PVDF polymer of 0.2 W·m^−1^·K^−1^. This increase in thermal conductivity can be attributed to the presence of CNT in the polymer matrix. Due to high Van der Waals interactions, CNTs have a tendency to form aggregates and weak interactions between the polymer and MWNTs are expected. However, these interactions between MWNTs can be overcome through the proper functionalisation of CNTs with an appropriate polymer, such as PVP, and thus, the overall thermal conductivity of the polymer nanocomposite can be increased.

The molecular weights of PVP used in this study are 10,000, 40,000 and 55,000 g·mol^−1^. From the thermal conductivity graph in Figure 3, it can be confirmed that the presence of PVP has altered the dispersion of MWNT in the polymer matrix to yield an increase in the thermal conductivity. The higher polarity in PVP compared to the PVDF could induce strong π–π interactions with MWNT and as a result, could produce a more homogeneous composite due to the better dispersion of nanotubes. However, the difference in the molecular weight of these polymers plays a key role in the overall dispersion. One can observe from the result that PVP with a lower molecular weight of 10,000 g·mol^−1^ records a value of 3.64 W·m^−1^·K^−1^ at a weight percent of 3.38%, which is a 146% increase in the thermal conductivity from the non-functionalised MWNT/PVDF composite.

PVP_40000_ and PVP_55000_ display their highest thermal conductivity values of 2.14 W·m^−1^·K^−1^ and 2.40 W·m^−1^·K^−1^ at a weight percent of 9.09% and 23.08%, respectively. Although they exhibit an increase in thermal conductivity of about 44.6% and 62.1% compared to pure PVDF, they are comparatively lower than the highest thermal conductivity observed for PVP_10000_. This difference observed with the same type of polymer and same structure, with only differences in molecular weight, could be attributed to factors like wrapping behaviour, polymer structure and the geometric parameters of the constituents in the nanocomposites. PVP has amide bonds and pyrrolidone rings and they tend to possess a flexible backbone structure which leads to the polymer forming an interchain coil rather than a helical conformation [16,17]. Mu et al. reported that a substantially increased tensile modulus can be observed when the radius of gyration of the polymer is greater than the diameter of the high aspect ratio filler [18,19]. However, Davijani and Kumar reported that the wrapping behaviour observed in few walled nanotubes (FWNT) and MWNT is different from SWNT due to the difference in their diameter, and as a result, helically ordered wrapping is only observed in SWNT [20].

Our thermal conductivity results illustrate that the presence of a functionalisation polymer with a low molecular weight produces high thermal conductivity in the composite at a low concentration. This could be attributed to the fact that PVP with high molecular weights possess longer polymer chains that wrap the nanotubes almost entirely. A thicker layer of polymer around the CNT due to higher surface coverage on the nanotube will still allow phonon transport from a nanotube through the layer of PVP to other nanotubes in the network, but this transport will be hindered, thus reducing the overall thermal conduction when compared to PVP with a low molecular weight. On the contrary, PVP with lower molecular weights and shorter polymer chains wrap the nanotubes with a comparatively thin layer of polymer to induce a proper dispersion without disrupting the phonon transport in the nanotubes, resulting in higher thermal conductivity.

### 3.3. Electrical Conductivity

The electrical conductivities of the PVP@MWNT/PVDF composite are illustrated in Figure 4, with three different molecular weights measured separately. The results exhibit electrical conductivities of 26 and 28.9 S·cm^−1^ for PVP_10000_ and PVP_40000_, respectively, at the concentrations that yield high thermal conductivity, confirming the presence of a conducting network in the composite at these concentrations. However, in the case of PVP_55000_, a weak conductive network has been witnessed, owing to the fact that the long polymer chain could have wrapped more than a single MWNT together or a single nanotube wrapped entirely, leading to higher surface coverage, thus compromising the formation of a conductive network. This could have hindered the transport of electrons in the overall composite, yielding a comparatively low electrical conductivity of 1.09 S·cm^−1^ which continues to decrease with an increasing concentration of PVP until a threshold point is reached at a loading of 16.67 wt %, beyond which no relative change in electrical conductivity is observed. 

### 3.4. Differential Scanning Calorimetry

The degree of crystallinity exhibits an apparent influence over the thermal conductivity of polymer-based nanocomposites [21,22]. Furthermore, it is a known fact that CNTs exhibit an excellent nucleation effect for the crystallisation of semicrystalline polymers [23]. W.-b Zhang et al. found that the crystal form of PVDF does not change when in a CNT–PVDF or PVP functionalised CNT–PVDF nanocomposite, confirming that the enhancement of thermal conductivity and variation of crystallinity are not induced by the change in crystal form of PVDF, but are dependent on the contents of CNTs in the polymer matrix [24].

Figure 5 shows a set of DSC curves for PVP_10000_ at various concentrations. The curves for the other molecular weight polymers are provided in the Appendix A (see Appendix A provide detailed summaries of the properties observed, Appendix A). Figure 6 provides the degree of crystallinity observed for each composite as a function of concentration for each molecular weight of the wrapping polymer. The DSC results obtained for the three types of PVP mimic their respective thermal conductivity results. PVP_10000_ exhibits a better thermal conductivity at a concentration range of 2.44 wt % and also displays a high degree of crystallisation of about 42.78%. Similarly, a high degree of crystallisation is observed in PVP_40000_ and PVP_55000_ at a concentration range of 16.67 wt %, but a comparatively lower thermal conductivity is displayed. This is due to the fact that in these concentration ranges of PVP, the MWNT, in addition to being well dispersed, also exhibits a conductive network in the PVDF matrix, thus allowing a high thermal conductivity and a higher degree of crystallisation. 

### 3.5. SEM

Nanocomposites prepared with non-functionalised MWNT in a PVDF matrix exhibit poor dispersion. From the SEM image in Figure 7a, severe agglomeration of nanotubes can be seen and is highlighted in the image by red circles. Due to high Van der Waals interactions, the nanotubes present are clustered together in the form of aggregates. 

However, in the PVP functionalised MWNT/PVDF samples of the same composition, unlike non-functionalised samples, an absence of clusters can be observed. This is due to the presence of PVP polymer in the composite, which adsorbs into the walls of the CNTs, thus decreasing the interfacial interaction. This distribution of MWNT in the PVDF matrix can be witnessed in all three molecular weights of PVP, confirming that the dispersion effect of MWNT in the PVDF matrix is highly improved with the wrapping of PVP, irrespective of the length of the polymer chain. A similar observation can be witnessed in the fractured SEM images. Figure 7e shows no clear presence of nanotubes, unlike the other images, which could be due to the cause of aggregation.

### 3.6. PVP vs. P4VP

To better understand the role of the polymer structure in non-covalent functionalisation, two polymers of different structures (see Figure 1) with similar molecular weights were chosen to make a composite and the thermal conductivity was measured (Figure 8). PVP 55,000 g·mol^−1^ and P4VP 60,000 g·mol^−1^ were used to functionalise the MWNT. P4VP at a weight percent of 37.5% exhibits a high thermal conductivity of 2.79 W·m^−1^·K^−1^, which is an increase of 89.1% compared to the sample without MWNT functionalisation. The polymer structure of P4VP with the presence of a pyridine group in the side chain is different to PVP [25,26]. The behaviour in which a polymer is wrapped onto a nanotube plays a key role in achieving a better dispersion in a polymer matrix and only a better dispersion leads to a higher degree of crystallisation and eventually enhances the thermal conductivity of the overall composite [27]. It is a known fact that non–covalent functionalisation happens through π–π stacking. In the PVP functionalisation of nanotubes, the π system in the carbonyl group of PVP and the π electron in the carbon nanotube engage in a π–π interaction. However, in the case of P4VP, the π bond in the pyridine ring and the π electron engage in a π–π interaction. Higher molecular weights of P4VP lead to a higher number of pyridine rings and eventually, more π bonds. This could have initiated a different wrapping behaviour in P4VP functionalisation, and maybe instead of helical wrapping, adsorption parallel to the length of the nanotube was achieved, eventually leading to a better dispersion at a different concentration than PVP of a similar molecular weight. This confirms that the polymer structure plays a key role in wrapping behaviour and dispersion capabilities.

## 4. Conclusions

In summary, a comparison of pristine MWNT/PVDF composites and non-covalently functionalised MWNT/PVDF composites with different molecular weights has been analysed. PVP and P4VP have been used as the functionalisation polymers for CNTs. The thermal conductivity measurement of the nanocomposites shows an overall increase. The molecular weight of the polymer used in the functionalisation of MWNTs has a significant impact on the thermal conductivity of the composite. 

The dispersion of MWNTs in the polymer matrix is improved with all three molecular weights of PVP, but the thermal conductivity shows a better response with low molecular weights due to the fact that short polymer chains wrap the nanotubes, improving the dispersion without compromising the connectivity of the nanotube network. The molecular weight of the polymer is not the only factor influencing the thermal and electrical property of the composite. The comparison data for PVP and P4VP confirms that the structure of the polymer and their interaction with the solvent also play a key role. 

## Figures and Tables

**Figure 1 polymers-11-00162-f001:**
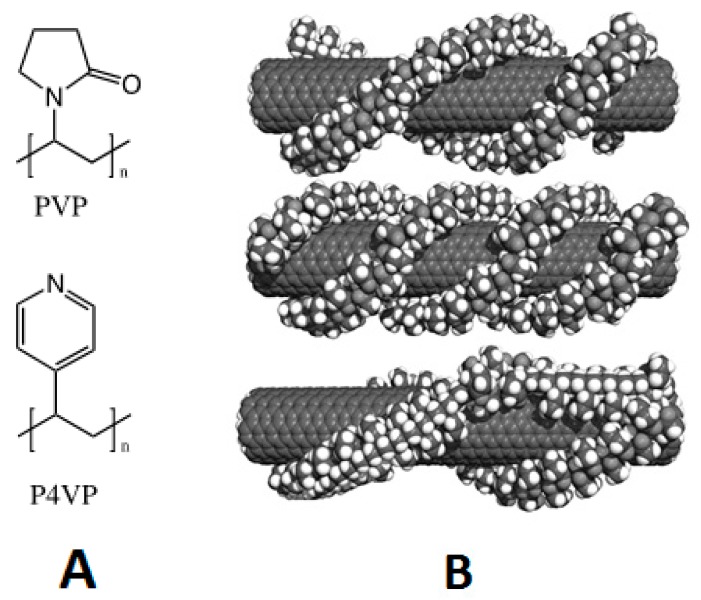
(**A**) Structure of PVP and P4VP; (**B**) computer model of PVP wrapping arrangement on a carbon nanotube [15].

**Figure 2 polymers-11-00162-f002:**
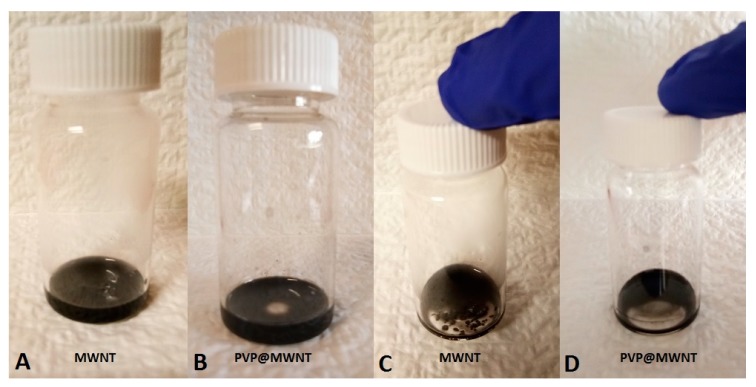
Optical image showing dispersion of MWNT and PVP treated MWNT in DMF: After being treated with sonication (**A**,**B**) and after being statically placed for 120 h (**C**) and 240 h (**D**).

**Figure 3 polymers-11-00162-f003:**
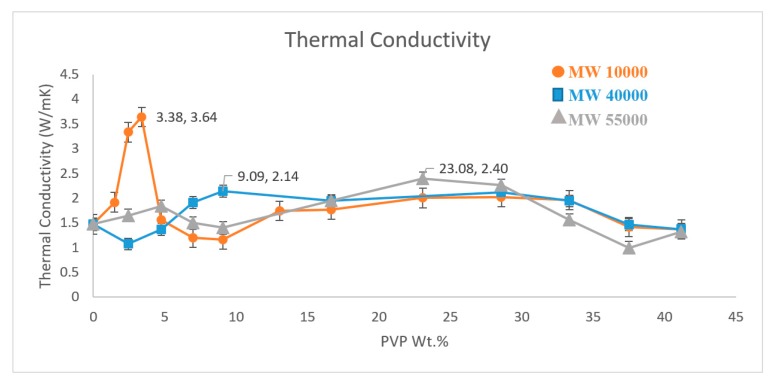
Thermal conductivity measure of three different PVP functionalised MWNT/PVDF composites.

**Figure 4 polymers-11-00162-f004:**
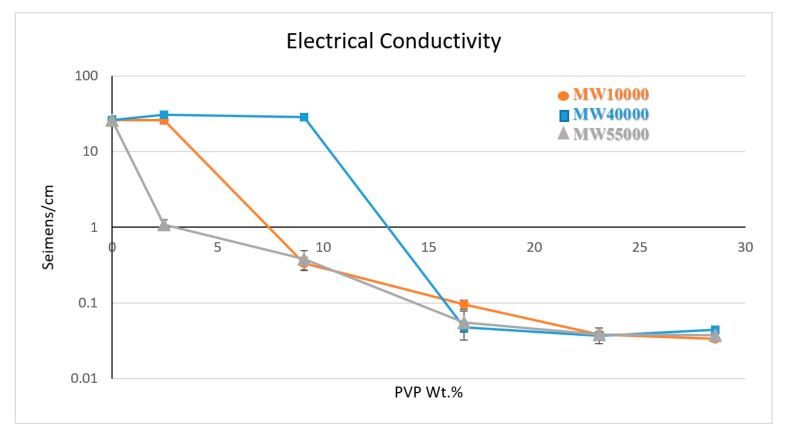
Electrical conductivity measure of three different PVP functionalised MWNT/PVDF composites.

**Figure 5 polymers-11-00162-f005:**
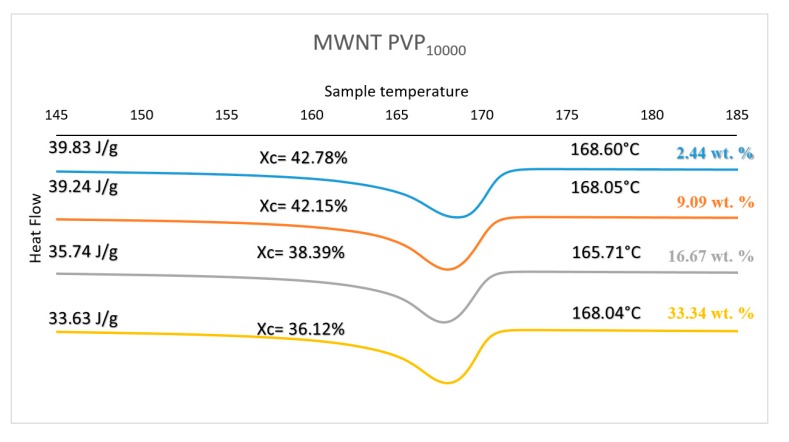
Thermal Properties of PVP_10000_ functionalised MWNT/PVDF composite.

**Figure 6 polymers-11-00162-f006:**
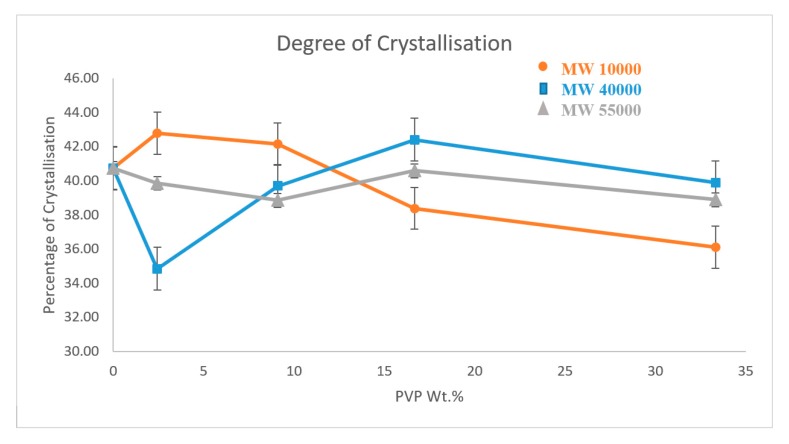
Crystallisation vs. wt % of PVP@MWNT/PVDF composite.

**Figure 7 polymers-11-00162-f007:**
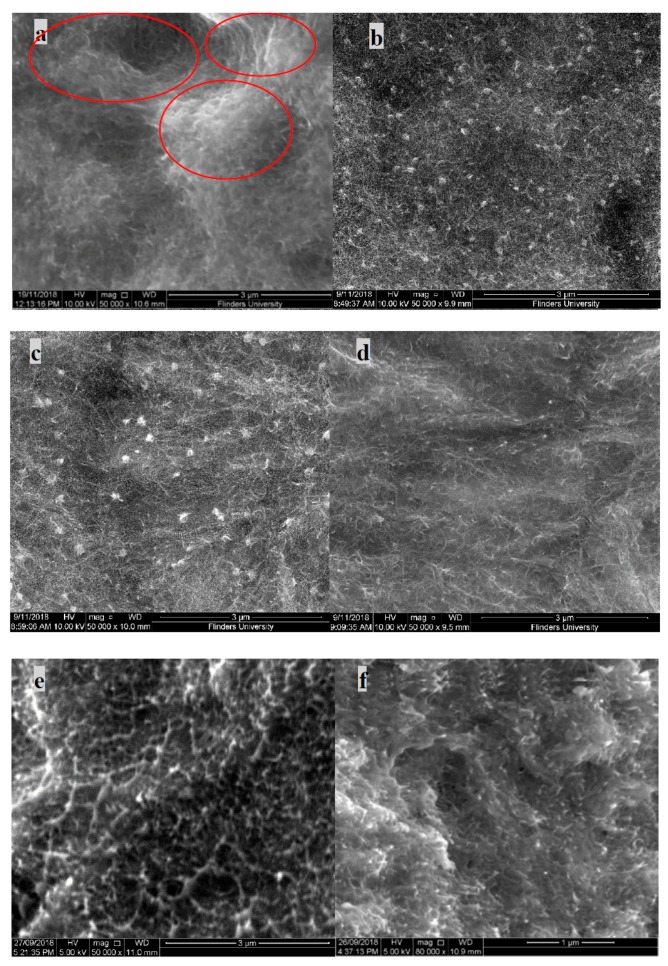
SEM image (Anterior) of MWNT/PVDF composite: (**a**) PVP@MWNT/PVDF composites; (**b**) PVP_10000_ at 2.44 wt %; (**c**) PVP_40000_ at 9.09 wt %; (**d**) PVP_55000_ at 23.08 wt %, SEM image (fractured side view) of MWNT/PVDF composite; (**e**) PVP@MWNT/PVDF composites; (**f**) PVP_10000_ at 2.44 wt %; (**g**) PVP_40000_ at 9.09 wt % and (**h**) PVP_55000_ at 23.08 wt %.

**Figure 8 polymers-11-00162-f008:**
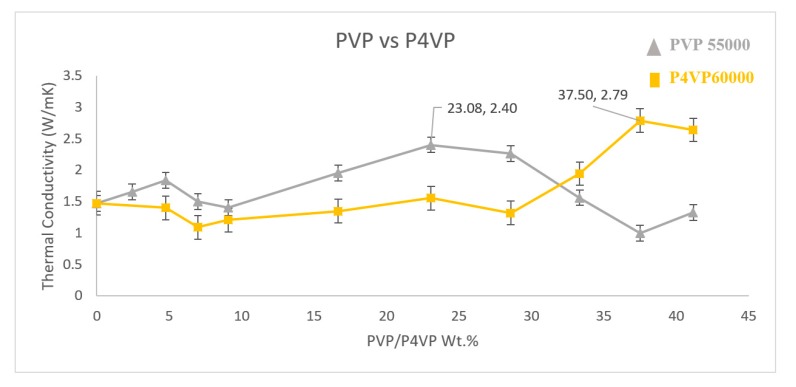
Thermal conductivity measure of PVP and P4VP functionalised MWNT/PVDF composite.

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
