# Peer review of "Role of Molecular Weight in Polymer Wrapping and Dispersion of MWNT in a PVDF Matrix"

_polymers, 2019, doi:10.3390/polym11010162_

Round 1

Reviewer 1 Report

Title: “Role of Molecular weight in Polymer Wrapping and 3 Dispersion of MWNT in a PVDF Matrix”

Authors: Muthuraman Namasivayam, Mats R Andersson, Joseph Shapter

polymers-408365

The paper “Role of Molecular weight in Polymer Wrapping and 3 Dispersion of MWNT in a PVDF Matrix” is well written, figures are legible, comprehensive data are supported with adequate literature. Because of that I recommend this paper for publication in Polymers.

Small remarks:

1/ Description of Fig. 1 should be clarified because contain also structures of polymers employed in this work.

2/ On Figure 4 temperature as well as concentration should be used in the same precision

3/ Units should not be combined with numbers

Author Response

See file

Reviewer 2 Report

The subject is of interest to the general readership of the journal; however, extensive rewriting and additional work is required, which is beyond that of a revision. Moreover, I feel that the Authors summarized the observations and did not address necessary explanations which weaken the scientific insight of this study. I am recommending a rejection based on the detail comments as below:

In the introduction section, the authors didn’t cite a single work from the available previous CNT/PVDF literatures. Please include recent works and compare.

Addition to the comment 1, it is hard to understand what is new in this work compared to those of already available in the scholarly works. What is the novelty of this work? Please explain clearly in the text.

“We study the extent of polymer wrapping at each concentration …” – Authors didn’t quantify the wrapping of the polymer. Extent of polymer wrapping on CNT for each polymer concentration was not quantitatively compared. Without this characterization, the scientific aspect of this work remains very weak. Please include.

What is the particular application of MWNT/PVDF composite studied in this work? Authors mention the applications of CNT which are very well know, however, didn’t talk about the expected use of fabricated composite. Please include.  

“All the carbon nanotubes used in this research were MWNTs purchased from Sigma Aldrich…” – Mention the number of walls and % of impurities.

“The first step was to obtain the “solution” containing PVP or P4VP wrapped MWNTs…” – At this stage of the experiment, how did the authors confirm that the MWCNT’s were truly wrapped by the polymer? Please characterize this suspension and provide concrete experimental evidence to support your statement.

“First step: 1 mg of unmodified MWNT was dispersed in 250 μL of DMF…” – However, it is known that unmodified CNTs can’t be “dispersed” in DMF and the concentration selected here is also not low. What is the evidence that the CNT was fully dispersed?

“… and sonicated for 10 minutes.” – Please provide the details of the sonicator (make, power and so on). Based on what/how did the authors optimize the sonication time of 10 min?

“Then, a defined amount of PVP or P4VP polymer were dispersed in the CNT solution. The mixture was sonicated for a period of 45 minutes and then left undisturbed overnight.” – In general, it is known that such a short time of sonication is not enough to make a “CNT solution” in DMF. In many cases, it would be a suspension and CNT will precipitate if left undisturbed. Please clarify.

 Secondly, based on what/how did the authors optimize the sonication time of 45 min at this stage? Did the authors expect and studied a length shortening of the CNT due to this treatment (this would depends of sonication power and CNT impurities)?

If the CNT after the 1st step as described in section 2.1 is dispersed then it would be stable without phase separation for prolonged period of time. See: Polymer 100 (2016) 244-258, fig. S3e) as a reference on CNT dispersion. Please include experimental evidence on CNT dispersion.

“Second step: The mixture is mixed with 20 mg of PVDF polymer in 200 μL of DMF and sonicated for a period of 4 hours.” – Similar to the previous comment, how did the authors optimize the sonication time of 4 hrs. and what was its effect on CNTs?

“Samples were imaged from both the anterior and fractured side view.” – Which views are presented in fig. 6?

Without a continuous line and different colors and symbol shapes it is hard to follow Fig. 2 data. Moreover, B/W print would be very confusing. Similar changes are needed for Fig. 3, 5 and 7. Please revise.  

In the figures mentioned in comment 12 and in the corresponding text, reword “Concentration (mg)”. Unit should not be in mg and therefore is confusing.

What does it mean by the “Thermal Conductivity PVP” as a title in Fig. 3?

“Higher polarity in PVP compared to the PVDF could induce a stronger π-π interactions with MWNT and as a result could have a more stable composite.” – What does “more stable” composite mean here? How does composite stability affect the thermal conductivity?

The explanation provided in line 137-143 is not clear while discussing the reason why PVP10000  shows high conductivity.  Please clarify.

“This could be attributed to the fact that PVP with high molecular weights possess longer polymer chains that wrap the nanotubes almost entirely. A thicker layer of polymer around the CNT will still allow phonon transport…” – How does the longer polymer chains that wrap the nanotubes can essentially make the coating layer thicker? How is the thickness of this coating (in z direction) related to the length of the polymer chain?  

“The results illustrates that the presence of a functionalisation polymer with low molecular weight tend to exhibit high thermal conductivity in the composite at low concentration.” – Why? Please explain this effect of lower concentration of low molecular weight PVP on the high thermal conductivity of the composite.

“…owing to the fact that the long polymer chain could have wrapped more than a single MWNT together or formed a thick wrapping around a single nanotube, thus compromising the formation of a conductive network.” – This is hard to imagine. Can you please prove this phenomenon using TEM or any other characterization method?

“… which continues to decrease with increasing concentration of PVP until a threshold point is reached at 0.2 mg beyond which no relative change in electrical conductivity is observed.” – How can the authors explain this effect? What happens/ is the mechanism when PVP is 0.2 mg?

“… also exhibits a conductive network in the PVDF matrix thus allowing a high thermal conductivity and higher degree of crystallisation.” – It is not clear how does the conductive network help to have higher degree of crystallinity. Please explain.

SEM images provided in this study in fig. 6 are not conclusive at all. “A comparatively higher distribution of MWNT throughout the samples” as claimed by the authors does not look convincing. A fracture surface SEM (see comment 12) could be more interesting. A reference of SEM images of CNT dispersion in fractured polymer composite can be found in Polymer 100 (2016) 259-274.

“… confirming that the dispersion effect of MWNT in PVDF matrix is highly improved with the wrapping of PVP irrespective of the length of polymer chain.” – None of the characterizations provided in this work definitely show wrapping of PVP around CNT. It is strongly suggested that the authors show TEM images or any other method as acceptable evidence of wrapping.   

“Unlike PVP, P4VP is hydrophobic in nature and could disperse well in acidic aqueous solution [18, 19].” – How is this statement relevant to the current study? Further explanation provided in line 219-226 are not clear as well. Please revise.

How is the electrical conductivity of P4VP containing composite if it is compared with the data presented in fig. 3?

Please include the mechanical characterizations of the composites studied in this work.

English language of this manuscript needs improvement (spacing, spelling, sentence construction, typo, grammar). I recommend the authors to proofread their article, or to use a language editing service. Some examples are:

functionalization

a stronger π-π interactions

a defined amount of PVP or P4VP polymer were

crystallisation

Many more...

Author Response

See file

Reviewer 3 Report

The paper showed the influence of MW of PVP on CNT dispersions and wrapping effects.

The paper showed high scientific merits. The only concern i have is, can the authors provide direct interfacial interaction parameters between the polymers and CNT? for example, can the materials of Polymer/CNT hybrid be collected and tested for interfacial strength? The crystallinity degree, wrapping layer thickness, dispersion quality etc. should influence their mechanical or thermal response and should be examined. 

Author Response

See file

Round 2

Reviewer 2 Report

I would like to request the Authors to upload the manuscript showing the revisions using the track changes or a different color. From current submission it is difficult to see the revision and compare with the older version. Thank you!

Author Response

Please see attached file the responses to your comments.

Round 3

Reviewer 2 Report

Acceptance is recommended upon English language correction. Some examples:

- Electrical and thermal conductivity of a polymer composite depends ...

- Much work towards understanding the effect...

- PVDF with melt flow rate (MFR)....were used...

- The nanocomposites are prepared through solution...

- The mixture is mixed with 20 mg of PVDF polymer...

- DSC results obtained for three types of PVP mimics....

- There are many more. Please correct.